# Privacy-Preserving Inference on the Edge: Mitigating a New Threat Model

## ABSTRACT

With the explosion of machine learning at the edge, there's a major need for privacy-preserving machine learning for edge devices. As the number of devices in homes and other private spaces increases, we can expect to see more malicious actors exploiting the inherent recording capabilities in these systems to harm people. This paper proposes that machine learning is not the problem, but rather a solution, which along with the use of a secure enclave, offers a pragmatic approach to preserving privacy. With a simple programming framework, we show how machine learning application developers can be as productive as usual, while still keeping user data private. We demonstrate our implementation for privacy-preserving machine learning on an embedded system, the Nordic NRF5340 PDK with Arm Cotex-M-33, using a relatively large model for person-detection.

## KEYWORDS

privacy, security, TEE, edge inference, secure enclave, embedded systems

ACM Reference Format:

Anonymous Author(s). 2020. Privacy-Preserving Inference on the Edge: Mitigating a New Threat Model. In *Proceedings of First International Research Symposium on Tiny Machine Learning (tinyML) (TinyML '21)*. ACM, New York, NY, USA, 9 pages. https://doi.org/10.1145/nnnnnnn.nnnnnnn

## 1 INTRODUCTION

Always on sensors are widespread in the everyday devices we all use from smart speakers (e.g Alexa) to home security cameras (e.g Nest) etc. They have the potential to provide a variety of useful features ranging from ubiquitous computing to automated security to in-home health and activity monitoring. Furthermore, many more consumer products are expected to include always on image or sound sensors as an incidental part of their operation, for example so an oven can respond to a voice interface, or a TV can power down when no person is nearby. These applications work by piping the output of image and audio sensors into machine learning models that run locally on the edge device complemented with more sophisticated inference running in the cloud.

However, in spite of their usefulness, there is growing consumer concern around the proliferation of such devices and the potential for both inadvertent and/or malicious threats that can leak sensitive

and private information. Historically, such concerns have been addressed in recording devices by adding a mute button, and having visual feedback like a LED to indicate whether a camera or microphone is on or off. A typical home may contain tens or hundreds of these devices in the near future, so the traditional approach of a recording indicator is unsustainable.

These indicators and controls exist because users care deeply about their conversations and activities being recorded. Recordings can be highly embarrassing or even put people in danger if they're shared, as we'll discuss below. With current general-purpose computing devices, there's no easy way to prove that audio or image sensors are only used for ephemeral purposes, and that no recording is possible.

This is a dilemma for innovators, since there are a lot of important medical, environmental, and consumer products that would benefit from applying machine learning to this kind of data, but have no interest in recording the raw sensor output. The most likely future with the current status quo is that scandals occur and any device with a recording capability becomes heavily restricted and distrusted.

This paper proposes an approach to address some of the threats with always on sensors and alleviate consumer privacy concerns. Our approach is to treat the access to raw recorded audio, video, or other sensitive data by malicious actors as a threat model for security engineering. We want to allow blessed machine learning models to read the raw data and output the intended non-sensitive information ("Is there a person in front of the TV?", "Did someone say a wake word?"), while ensuring that even with physical access to the device, it's not possible for any other parts of the system to capture and store the images or audio, or even derived information, like a facial or voice signature that might identify an individual.

The key intuition behind our idea is to recognize that for a large number of such consumer products, the basic inference building blocks can be reduced to a few key wakeup primitives that kickstart a more sophisticated inference pipeline. For example, most smart audio devices rely on detecting a wakeword, many visual systems rely on face id, and many indoor sensing systems rely on primitives like detecting motion or counting the number of humans. If instead of securing the whole inference pipeline, we could securely run the inference needed for these wakeup primitives, we could ensure that the more complex ML models only get access to processed features when the wake up action triggers.

To secure these wake up primitives, we leverage recent advances in secure enclaves that have historically been used to secure keys. By keeping the raw sensor data in a part of the system inaccessible to applications and only allowing limited results from machine learning models to be exposed, we can protect users from the threat of recording while allowing audio and video sensors to be used in benign ways.

## 2 THREAT MODEL

The primary issue is that recording and storing raw, sensitive data can expose unwanted insights about a specific individual or party. In this section, we outline several possible ways that this unfiltered data can be revealed to malicious actors.

### 2.1 Device Vulnerabilities

Security vulnerabilities are especially common in edge IoT devices, given the field's strong focus on time to market. Combined with the fact that these edge device are packed with sensors and often store a significant amount of data, it's no surprise that these devices are an enticing target for attackers.

*2.1.1 Your Toaster is Listening.* Acme Corporation introduces a new smart toaster that listens out for smoke alarm beeps and shuts off the heating elements automatically. It's also connected to the internet so you can start your toast cooking from bed before you get up.

The Democratic Republic of Ruritania's Secret Police find a flaw in the web control panel that the toaster exposes, and are able to upload arbitrary software to the device. They decide to use this against foreign critics of the regime, and are able to transmit recordings of all their kitchen conversations using the microphone that was installed to detect smoke alarms.

*2.1.2 Analysis.* Devices are vulnerable to this attack simply due to the fact that there's no strong mechanism for protecting sensor data from unauthorized access. In addition, by having the ability to store microphone data, the toaster becomes a target for attacks. Turning the application ephemeral, by not storing any audio data, would effectively mitigate this sort of attack, no matter what vulnerabilities may be present on the device. The module that has access to the sensor data cannot store it, and the rest of the system can't even access the raw data.

### 2.2 Nefarious Application

Intuitively, people are worried about devices recording them in detail, but don't have similar concerns about sensors detecting less detailed information about the environment. However, a use case that's accepted and perceived as benign can be changed into something unacceptable.

*2.2.1 Ads that Watch You.* A large US city installs video billboards from a startup in its bus shelters. To avoid light pollution and reduce energy usage, these billboards use a low resolution image sensor to detect when people are nearby, and only light up when they might be seen. They all have an internet connection to update software and advertising content.

The startup decides to supplement its advertising income by streaming video from all of the image sensors and selling live access to anyone willing to pay, including private investigators and foreign governments. This can be enabled by a simple software update.

*2.2.2 Analysis.* The main source of this threat is the fact that the application has unrestricted access to raw sensor data. Although the initial intention was to only use the image data to detect people, since the application has full access to the image sensor, the application can be re-purposed to stream the raw data as well. Along

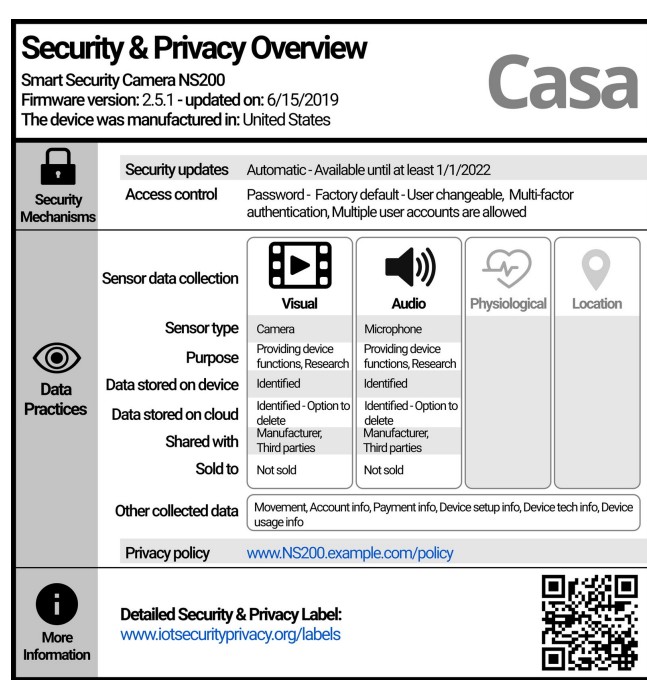

**Figure 1: Similar to nutrition labels on food, a label can be placed on IoT devices describing the onboard sensors and how data is used. [5]**

with the previous threat, this further highlights the importance of isolating the application from the data.

### 2.3 User Consent Model

Consumers of edge devices have limited insight into how the sensor data is being used, which prevents them from being able to understand and trust the device. Simply stating that a sensor is present on a device has no value, and can lead to consumers ignoring all such warnings.

*2.3.1 Prop. 65 for Edge Devices.* In light of recent scandals around edge devices with sensors, legislation is passed that forces all products to add a label that explicitly states every sensor contained within the device. As a result, consumers begin to ignore these noisy labels, which provide little insight into how sensor data is actually used.

A large e-commerce company develops a smart home device for controlling lights with just a clap, and in accordance with legislation, warns consumers that the device consists of a microphone. Unbeknownst to consumers, the microphone is also listening to conversations in an attempt to target users with personalized ads.

*2.3.2 Analysis.* Similar to the permission model on mobile devices, a system needs to be in place to ensure that users can understand and make decisions about the sensor data that an application can use. However, a key weakness of the on-mobile application permissions model is that users typically end up approving all permissions without really understanding the reason, which defeats the entire purpose.

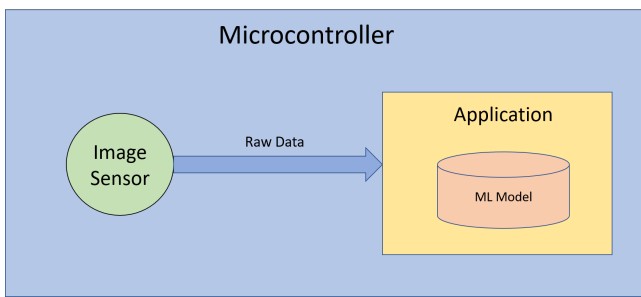

**Figure 2: Traditional Implementation**

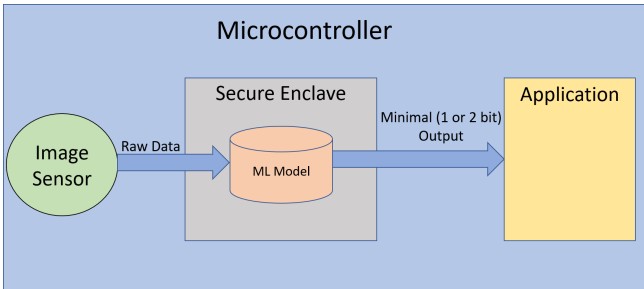

**Figure 3: Proposed Implementation**

Another approach that has gained traction recently is the use of privacy "nutrition labels", as shown in Figure 1. Although it is a step in the right direction, there are major weaknesses. If the label is too detailed, users may no longer pay attention them. On the other hand, if the label is not detailed enough, nearly every edge device will have the same label, effectively rendering the label useless. Achieving the right balance between detail and noise is non-trivial, and as a result, this approach may not be the most effective. In general, the user only wants the sensor to detect information about the environment, not record them, so therefore it's crucial that a clear consent model helps users really understand the amount of information that a device can access.

### 2.4 Mitigation

The 3 threats presented in this section reveal the key attributes that would make up the ideal edge device. First, an attacker with control of a device should have no way to access raw sensor data. Second, the application itself should have no access to raw sensor data. Finally, the consumer has the ability to understand how sensor data is being used and control how an application can use sensors. With these properties, the ideal edge device is able to effectively perform the same tasks as a traditional device, while also preserving privacy.

### 3 PRIVACY-PRESERVING EDGE DEVICES

It has not escaped our notice that this threat model applies equally to legacy devices not currently performing machine learning completely on-device or on the edge. For example, a legacy surveillance camera with no model for facial recognition can still record private footage. We have chosen to re-explore this problem now not because machine learning exacerbates the threat but because it offers a solution: the removal of the human from the loop.

Removing the human from the loop suggests that the integration of these black-boxed neural networks into sensors is a great step forward towards privacy-preserving machine learning. With this in mind, we present our abstraction for these privacy-preserving smart sensors: auditable machine learning models integrated into sensor hardware with a secure enclave.

In this abstraction, the manufacturers of these smart sensors are a trusted but auditable party; in contrast, the application developers who use these smart sensors are considered untrusted parties. Typically, application developers are responsible for writing the firmware to interface with sensors, store the resulting data, and process the data to generate insights. Integrating the machine learning model co-opts the above process by using a trusted driver to interface with the sensor, a secure enclave to temporarily store input data, and a DNN to process data and generate a filtered output. By removing the human from the loop, these sensor modules can output low-dimensional or even binarized outputs and rate-limit how often outputs are generated. Although hardware attacks are likely against our solution, we limit the scope of our attacks to those equivalent in effort to an attacker simply including an additional malicious sensor. In other words, we do not protect against an attacker adding an additional bugged microphone or camera; we instead ensure that the attacker does not find a way to compromise our existing hardware at scale.

### 3.1 Machine Learning as a Solution

The use of cameras has traditionally relied on a human being in the loop to process and synthesize the raw video stream into actionable insights. Replacing the human's role in processing data with a machine learning model helps privacy by ensuring that only the camera itself has access to the raw input. In the past, legacy devices had no option but to record and store all its data because manual inspection by a human worker outperformed any computation-based solution. Now, with edge-deployable machine learning models capable of performing tasks like person detection with sufficient accuracy, it has become possible to remove the human from the loop and rely on the model for processing. As raw sensor data is fed into a machine learning model, the data undergoes a non-invertible transformation into a task-specific result.

However, the simple application of machine learning is not a catch-all solution to the problem. For one, the output of the machine learning model may have enough information for an attacker to reverse-engineer the input fed into the model [6]. Therefore, we reduce the dimensionality of the output, for instance to a single on/off bit, to limit the exploitable information that an attacker can use to invert the output to an input. With this approach, applications have no access to raw sensor data, but rather only deal with a 1 bit output of the machine learning model (e.g. is a person detected?). If we combine this with rate-limiting over time, we can ensure too little information is revealed to reconstruct the original input. As a result, a malicious application is only provided the bare minimum data required, and thus has no ability to recreate and record or stream out raw sensor data.

## 3.2 Trusted Execution Environment

To secure sensitive data, a trusted execution environment (TEE) is necessary to ensure that applications have no way to directly access sensor data. A system with a TEE has two distinct areas: a secure world and a nonsecure world. This paper focuses on ARM TrustZone, which offers strong memory protection between the secure and nonsecure worlds. In particular, the secure world has access to resources in both the secure and nonsecure world, while the nonsecure world has no access to the secure world.

In this system, applications are part of the nonsecure world, while sensor data and machine learning models are part of secure world. Applications may only request a inference from the secure world, which returns a minimal output from the model back to the nonsecure world. With this hardware-level protection in place, even if an application were to be compromised, the attacker would have no access to any sensors.

## 3.3 Hardware Separation

In order to use the TEE effectively, the sensor output must directly feed into the secure memory region, making the sensor inaccessible from applications. This restriction means that the sensor can only be used for machine learning applications, which could pose a problem to some devices which also use the same sensor for general purpose applications. For example, a device may contain a camera that is used both for person detection as well as for video conferencing. In this case, devices must contain duplicate sensors, one that is locked down to the TEE and used exclusively for machine learning, and one that can be used for general purpose.

## 3.4 Auditability

As described earlier, the primary concern is with recording more data than required by the machine learning model. Temporary storage of a few seconds of audio data is no problem but storing several minutes worth of raw audio is unnecessary and risky. In lieu of a formal and theoretically rigorous approach to guaranteeing the security of our system, we offer a pragmatic way to audit the data recording and evaluate the security of our system.

The simplest approach to preventing the recording of significant amounts of sensor data is by physically limiting the memory available. Unlike a traditional system, where the sensor and the application share the same memory space, in this system, the two memory systems can be sized independently of each other.

In addition to simply limiting the size of sensor subsystem memory, the data recording can also be audited by filling up memory with special data values, similar to the idea of stack canaries. Memory space used by the sensor for recording will lose the initial value, whereas memory space that was unused will retain the initial value. This simple approach allows for auditing memory usage, and any excessive memory consumption would signal that the sensor is recording more sensitive data than required.

To enforce the restriction that all incoming data must be contained within its allocated arena, we further require that the checksum of the memory holding the model's weight values and the checksum of the memory holding the application code must match the manufacturer's initial checksum for both. This prevents an attacker from simply storing raw input out of the arena and overwriting the values of the model or the program itself.

## 3.5 Consent Model

The final part of this privacy-preserving system is a simple and straightforward user consent model. Even with all the described hardware support in place, consumers need to be able to understand how much information about the environment devices will be able to gain. Even if raw sensor data is strictly protected and only a single bit output is available, an application may still be able to gain unwanted insights. For example, an application that can access the microphone every millisecond has significantly more information than an application that can access the microphone every second.

Therefore, an effective consent model not only informs users of what sensors are present on a device, but also the rate at which sensors are accessed. The secure world also enforces this rate, by only providing an output at the predetermined rate. This rate limiting policy is another element in preserving privacy. Not only are applications presented with a single bit output, but are also limited in how frequently this single bit output is produced. This places an absolute bound on how much information can be gained through sensors.

## 3.6 Scope of Protection

Although secure enclaves, and ARM TrustZone specifically, offer strong memory protection between the secure and nonsecure worlds, several side-channel attacks and more invasive exploits have been published against vendor-specific implementations of TrustZone [13]. Furthermore, academia has published papers on several orthogonal attack vectors that circumvent TrustZone's protections [8]. To this effect, we offer a minimum level of protection in that an attacker cannot co-opt our framework to compromise machines at scale. In other words, we aim to ensure that attempting to circumvent our framework would be harder than an attacker simply adding an alternative listening or recording device. The maximum level of protection we offer is the maximum protection that TrustZone guarantees: memory protection between states.

In our framework, we offer application programmers the ability to integrate their application (nonsecure code) with the privacy-preserving ML runtime. Ultimately however, the hardware protections of TrustZone may be circumvented via the user application performing undefined behavior. Although this may not be systematically exploitable in a targeted manner, undefined behavior can ultimately reveal the secure image data. As such, we assume that the end user is not a malicious actor and will not attempt to cease operation of the entire device.

## 4 COMMON USE CASES

In order to support a wide variety of applications, this framework requires sensors to implement a few basic primitives, which can be thought of as the API interface to the secure world. These primitives only define the interface between the secure and nonsecure world, and are mostly decoupled from the actual model running in the secure world. In other words, two applications that are performing similar tasks but with different models (e.g. detecting a cat vs. detecting a banana) make use of the same primitives.

With this approach, nearly any visual or audio embedded machine learning application can be decomposed into a set of primitives, allowing for the same functionality while still preserving privacy. In this section, we look at three common applications and describe the primitives required by each of them.

## 4.1 Wake Word Detection

Wake word detection is one of the most common applications involving audio sensors on edge devices. In this application, a model continuously monitors audio data for a wake word, and on detection, triggers some other action within the application. The simplest application of this type only requires a primitive called `isWakeWordDetected`, which returns a boolean result of detection model. For applications with multiple wake words, a slight variant of this primitive is required, called `whichWakeWordDetected`. In this primitive, instead of returning a boolean, a low bit integer corresponding to the class detected by the model is returned instead. By utilizing a sensor with these two primitives, an application will be able to run nearly any wake word detection task.

## 4.2 Object Detection

Object detection is the visual analogue to wake word detection, and requires identical primitives. For simple applications, only the primitive `isObjectDetected` is required, which returns a boolean result indicating if the model detected the object it's been trained to detect. Similarly, for applications where multiple objects may be of interest, the primitive `whichObjectDetected` is required, which returns the low bit integer corresponding to the class that was detected.

## 4.3 Object Counting

Another common visual task is object counting, where the application requires a numerical count of the object of interest in a scene. The primitive `countObject` is required, which takes in an integer corresponding to one of the classes that the model is trained for, and returns an integer corresponding to the object count.

## 5 IMPLEMENTATION

## 5.1 Secure Enclave

To achieve the goal of performing inference while preventing the user application from accessing the original input data, we put the peripheral drivers, TFLite Micro runtime, and image buffer in the Secure World. This allowed us to prevent the Nonsecure World from accessing the input data from either the sensor itself or its SRAM storage and from probing the intermediate steps of the computation. However, the Nonsecure World code can still request the binarized output of the inference via an API call to give the application code flexibility in actually evaluating the model.

Although the ARMv8-M specifies some common features across all chips that implement the architecture, there is leeway for vendors to implement their own additions. This is the case with the microcontroller that was used to implement this project: the NRF5340 from Nordic Semiconductor. The NRF5340 specification specifically recommends disabling the architecture-specified Secure Attribution Unit (SAU) so that the NRF5340's System Protection Unit (SPU) can

be used in the way it is intended [12]. As a result, there are several chip-specific steps that need to performed to get have TrustZone perform as expected. We will describe the steps taken to do this below.

First, we divided the code base into nonsecure and secure source. This meant separate linker scripts, Makefiles, and startup code for each project. Since the NRF53 boots into secure mode, this meant that we needed to alter the startup code such that it would only initialize the hardware and set up the C runtime in the Secure World. In addition to the startup code, the NRF53 requires trusted code to set permissions for the RAM, Flash, and peripherals as needed for operation in the Secure and Nonsecure Worlds. The NRF53 provides 512 KB of SRAM comprising of 64 regions of 8KB each as well as 1024 KB of Flash comprising of 32 regions of 32KB each; the NRF53's SPU only offers region-level granularity. The NRF53 does not impose any explicit restriction on how many regions can be allocated to the Secure and Nonsecure Worlds, so there is relatively large flexibility in the size of each partition.

Next, the Secure World and Nonsecure World require separate stacks and interrupt handlers. The ARMv8-M family has the microprocessor boot in the Secure World. The ARM architecture specifically expects the handlers to start at address 0x0, so this means that the stack setup for the Nonsecure World must occur as part of the trusted code. Specifically, we defined a range of Flash and SRAM to the linker that the nonsecure code could be placed in. Afterwards, we set the Virtual Table Offset Register (VTOR) to shift the nonsecure stack and nonsecure interrupt vector table to the offset in Flash corresponding to the Nonsecure region. Once completed, we adjusted the Nonsecure Main Stack Pointer (MSP) to be at the start of the Nonsecure Flash region.

The steps up to this point are sufficient to completely isolate the Nonsecure World from the Secure World. However, assuming the application code runs in the Nonsecure World, there is a major limitation in that the Nonsecure World cannot trigger an inference from the Secure World. To solve this problem, we need to allow some functions implemented in the Secure World to be called by code in the Nonsecure World. The two requirements for this. First, the address that the Nonsecure code is attempting to branch to must contain a Secure Gateway (SG) instruction. Second, the instruction must be fully contained within memory that is part of the Secure region and marked Non-secure Callable (NSC). We thus define a special segment from one of the secure Flash regions as an NSC region so as to allow the function to execute in the Secure World. This needs to be done both at runtime (by marking said region as NSC) and at link-time.

Once the NSC region has been defined, we need to export out the function prototypes that the Nonsecure World can call; this is easily done by placing these function declarations in a separate header file that is then shared with the Nonsecure source. The function implementations must also be marked as being callable from the Nonsecure world using the cmse-nonsecure-call attribute; access to the Cortex-M Security Extensions (CMSE) API is provided as part of the GNU ARM compiler and will have the compiler generate the appropriate BLXNS and SG instructions to the Secure World and back. These marked functions are then exported out as an object file such that the Nonsecure source is aware of the existence of the NSC functions.

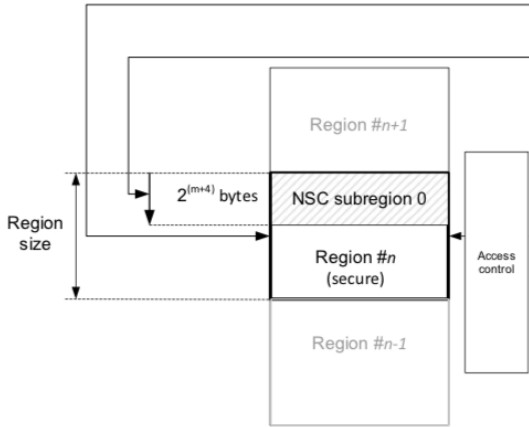

**Figure 4: NSC Region in Flash**

```
1  // Define a macro for Flash offset
2  // containing contents of Nonsecure World
3  #define FLASH_OFFSET 0x80000
4
5  // Check the processor's CPUID register
6  // to confirm architecture
7  assert(is_cortex_m33());
8
9  // Check that the processor is in secure state
10 assert(is_in_secure_state());
11
12 // Define sandbox flash offset into VTOR
13 SCB_NS->VTOR = FLASH_OFFSET;
14
15 // Configure security for flash
16 configure_flash_perms();
17
18 // Configure security for ram
19 configure_ram_perms();
20
21 // Configure security for gpio
22 configure_gpio_perms();
23
24 configure_nonsecure_gpio();
25
26 // Set the value of the NONSECURE MSP
27 uint32_t *vtor_ptr = (uint32_t *) FLASH_OFFSET;
28 __TZ_set_MSP_NS(vtor_ptr[0]);
29
30 // Set up the TFLite Micro runtime
31 setup();
32
33 // Define a pointer to a function
34 // in the Nonsecure World
35 nsfunc *ns_reset = (nsfunc*)(vtor_ptr[1]);
36 ns_reset = cmse_nsfptr_create(ns_reset);
37
38 // Branch to nonsecure reset vector one word away
39 // from secure Flash offset
40 ns_reset();
```

**Figure 5: TrustZone Setup in Secure World**

```
1  int main (void) {
2      // Request inference from Secure World 3x
3      for(int i = 0; i < 3; i++)  {
4          request_inference();
5      }
6
7    // Attempt to read from private image buffer
8    // Results in a fault
9    return *((uint32_t*)0x20000ecc);
10 }
```

**Figure 6: TrustZone Setup in Secure World**

A large caveat to this approach is that the programmer must take care to ensure there are no memory overlaps between the Nonsecure and Secure Worlds. This is required to ensure that the two resulting hex files (one for the secure output and the other for the non-secure output) can be easily combined before flashing the microprocessor. Due to ease of use, we used the srec-cat utility and the Intel hex file format. Once the hex files are combined, we use the onboard JLink programmer (interfaced with Nordic Semiconductor's nrfjprog utilty) to perform the actual programming operation.

### 5.2 Inference Engine and Model

The inference engine used is the Tensorflow Lite Micro runtime. The person detection model is the original MobileNet v1 network trained on Google's Visual Wake Words Dataset. The model consists of 14 depthwise separable convolution layers, an average-pool layer, a fully-connected layer, and a softmax. To reduce the needed computation, the model also eliminates 75% of the channels in each activation layer. The model weights were defined in a separate header file and loaded in as a BLOB as part of the Secure World.

All the complexity needed to deploy this ML model is abstracted away from the end user however. As the model, the data, and the implementation of the forward pass is in the Secure World, the entirety of the inference is done in the Secure World. The Nonsecure World, however, can access the binarized output of the inference by simply calling the accompanying function defined in the NSC API.

### 5.3 Hardware Setup

The model was trained using image data from a 2MP ArduCam module based on the OV2640, and ideally, we would have also used the ArduCam module for collecting images. However, the vendor-provided driver for the ArduCam did not function as expected. Instead, we collect images using the Coral Camera, a 5MP camera module based on the OV5645 image sensor, and downsample the frame. We concede that the use of a different-resolution camera may increase error, and given more time, we would try to fix the ArduCam driver, and directly communicate with it through the NRF53.

To interface with the Coral Camera, we use the Google Coral development board. This development board has a UART connection to the NRF53, allowing it to send frames captured from the Coral Camera. The software running on the Coral board is a simple Python script that uses the OpenCV library to capture a frame, downsample it, and then transmit it over UART, on a keyboard press.

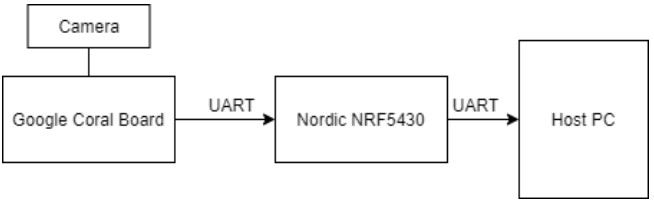

**Figure 7: Experimental Hardware Setup**

|  | No TrustZone | With TrustZone |
| --- | --- | --- |
| Branch to Secure | - | 86 |
| Inference | 581.022M | 581.108M |
| Branch to Nonsecure | - | 102 |

**Table 1: Execution time with and without ARM TrustZone, in cycles.**

All debugging output was done via a second UART connected to the host via a CP2102 USB-to-UART converter. A diagram of the experimental setup is shown in Figure 7.

## 6 RESULTS

### 6.1 Overhead

*6.1.1 Execution Time.* To calculate the execution time overhead of using TrustZone, we used the Cortex M33's DWT_CYCCNT register, which provides an exact cycle count. Table 1 summarizes the results.

*6.1.2 Code Footprint.* The additional code footprint is dependent only on number of secure functions that can be called from the nonsecure world, or in other words, the number of functions in the secure API. In addition, this footprint can be calculated analytically.

A function call to the secure world requires the SG instruction to branch into the secure world, and returning requires the BLXNS instruction, which results in an overhead of 8 bytes of instruction per secure world API.

The NSC region also introduces an additional overhead. Since the NSC region will consist of a branch to a function in the secure world, a single instruction, with a size of 4 bytes, is required per function. With a granularity of 32 bytes, the overhead introduced by this region is therefore $ceil(\frac{32}{4*n})$, where $n$ is the number of API functions.

Finally, the partitioning of code into secure and nonsecure may also introduce some overhead. Regions in flash can be marked as secure and nonsecure with a granularity of 32kB. This means that at the absolute worst case, an overhead of up to 64kB may be incurred. However, this is not a limitation of TrustZone, but rather of the specific implementation of this architecture.

### 6.2 Model Complexity

When we first started this work, we had a hypothesis that TrustZone would only be able to protect certain sizes of models and runtimes. However, we were surprised to see that even with a fairly large model (293 KB for a person detection model) and an overall large hex file (512 KB for model and runtime), the model still performed as we

expected. When we performed unit tests to access the image buffer in the Secure World from the Nonsecure World, we found that the Nordic faulted as we expected. Although we must further stress the limits of TrustZone with larger and more complex models, we see no evidence to suggest that model size would impact functionality as long as it can physically fit on-device.

## 7 PRIOR WORK

### 7.1 Input Protection

Prior literature has mostly focused on protecting the actual ML model itself. One interesting discussion has been centered on how to segment a large ML model such that certain layers performed in a secure enclave while others are performed outside of it [7]. However, we believe that to completely preserve privacy, we must place the input data, the model itself, and all scratchpad values inside the secure enclave to prevent an adversary from accessing intermediate values.

One promising approach to privacy-preserving inference lies in homomorphic encryption (whether it be partial or full). Microsoft Research has previously demonstrated that performing inference on encrypted data is feasible at relatively low latency for simple datasets such as the MNIST dataset [4]. However, fully homomorphic encryption suffers from inefficient evaluation of complex ML models; even if the bootstrapping process is optimized, the complexity of the operation is O(N$^2$) [2]. Partially homomorphic schemes offer improved efficiency but at the cost of not supporting some operations. This excludes some model architectures from being evaluated.

On the opposite end of the spectrum is the use of secure multiparty computation to shard work between multiple parties. Although this offers the advantage of distributing sensitive data across multiple parties without requiring encryption, communication speed bounds the latency of the operation. Furthermore, one common variant of secure multiparty computation, Beaver triples, enforces the use of fixed-point arithmetic [1]. Even more limiting is the requirement to only use linear operators; this severely limits the complexity of the models that can be evaluated in such a manner.

### 7.2 Software-Level Memory Protection

To protect memory at the software level, there are numerous systems approaches to do so in the literature. The Purify approach to doing so is to rewrite the target code's binary and instrument every load or store to memory with a function call to a state machine [9]. This state machine checked the status of memory and handled transitions in state appropriately. Alternative approaches to binary rewriting include protecting the software stack through use of stack canaries and redzones. By marking unused portions of the stack with a predetermined pattern (the stack canary), the trusted code can verify that the stack will not overflow [3]. The System V ABI also defines a different approach to stack canaries: redzones. These redzones are 128-byte memory regions defined after the stack pointer that will result in a panic of their contents are changed.

A more effective but costlier approach is to create a shadow memory that monitors the state of the memory space. This effectively requires a trap of every load/store instruction and an update of

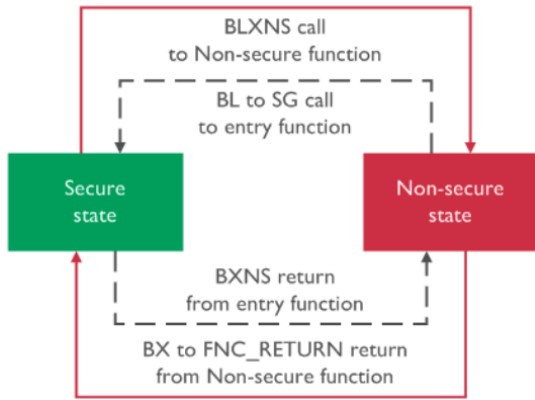

**Figure 8: Security State Transitions**

the shadow memory to monitor state. This enables a software-only approach to verifying the memory operations; however, this results in a drastic slowdown of the executing program. For this reason, hardware-level support for memory protection is a necessity for efficient edge ML operation.

Another approach to protecting memory is by using an operating system in a similar way as the TEE in our proposed implementation [10]. In this approach, the same mechanisms that protect kernel memory from being accessed by user applications also protect raw sensor data. Although this approach would have a similar effect as our work, the main drawback is the necessity of an operating system. Embedded systems at the edge are typically heavily resource constrained and are required to have a minimal latency, and as a result, may not be able to run an OS.

## 7.3 Hardware-Level Memory Protection

Hardware support for memory protection been a longstanding feature of the ARM microprocessor family. From as early as the ARMv6 family, variants of ARM microprocessors have supported the division of memory into Secure and Nonsecure Worlds via ARM TrustZone. However, only with the recent release of the Cortex-M23/M33 from the ARMv8-M family has ARM TrustZone found its way into the low-power Cortex-M family of microprocessors. TrustZone prescribes a set of hardware extensions that allow for the separation of systems software into a Secure and Nonsecure World, which presents an ideal setup for booting a Trusted Execution Environment (TEE) [11].

The core difference between the Cortex-A and Cortex-M implementations of TrustZone is the role of the secure monitor in transitioning between security states. The Cortex-A TrustZone requires a call to the secure monitor (via the SMC instruction) in order to switch states; in contrast, the Cortex-M TrustZone uses hardware extensions to perform the transition. This obviates the need for software to control the secure monitor. Although ARM TrustZone's primary use is to restrict software accesses from the Nonsecure World into the Secure World, it also provides additional protections to prevent invalid transitions into the Secure World.

## 8 FUTURE HARDWARE SUPPORT

Although this framework provides an effective way to preserve privacy, there are some limitations as a result of the underlying hardware. In particular, hardware support is required to ensure sensor data is restricted to the secure enclave, and a hardware-based manager in the secure enclave for models.

### 8.1 Connecting Sensors into Secure Enclave

A major limitation in the current implementation is that there's no hardware separation between sensors in the secure and nonsecure world. As previously discussed, restricting sensor data to the secure world allows for strong protection, as the raw data cannot be accessed by an application in the nonsecure world. However, commercial architectures today have no external interface for the secure world. This means that a sensor intended to be used exclusively in the secure world shares the same external interface as sensors used in the nonsecure world, and as a result, no hardware is present to restrict access to the sensor.

### 8.2 Model Manager

Secure enclaves are primarily designed for cryptographic keys, and as a result, have key managers for ease of use. Similarly, in order to make the secure enclave easier to use for machine learning models, a similar model manager is required. With this, models can be easily stored and updated through this interface, with minimal effort by a developer. In addition, this manager also ensures that models cannot be tampered with by any external application.

## 9 CONCLUSION

This paper presents a practical method for privacy-preserving machine learning at the edge. We show that through the use of a secure enclave, we can protect sensitive sensor data from attackers, and instead only expose a single bit output to the rest of the nonsecure world. In addition, we present a software-based method to audit any data collection, to ensure that only the bare minimum amount of sensor data is being recorded. Our framework effectively abstracts away the complexities of setting up ARM TrustZone while also providing maximum flexibility to the application developer. Our framework is also capable of working with complex and large machine learning models without any loss in security. For our next steps, we hope to add further security measures made available through TrustZone, such as hardware support for stack limits. In addition, we hope to add more software support for auditability through measures such as checksumming regions to verify integrity.

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
