# OpenReview forum: "Privacy-Preserving Inference on the Edge: Mitigating a New Threat Model"
_tinyml.org/tinyML/2021/Research_Symposium — tinyML 2021 Poster_

### Official Review · AnonReviewer3 · 2021-01-07

**Overall Merit Score:** 4

**Brief Summary:**

The paper presents an approach to protect privacy of sensor data which is acquired in small edge devices.
It uses the an MCU with Cortex-M3 processor and ARM Trustzone as target platform.
A "visual wakword" model is infered with TFLite Micro runtime  for demonstration.
The entire processing pipeline from sensor raw data to the classified input for a user application is put into the secure environment.

**Detailed Comments:**

This paper explores system aspects of trusted machine learning in deeply embedded edge devices.
In particular it demonstrates how machine learing itself can be used to increase data privacy with progressing digitalization.
This touches a very relevant matter to be discussed and evolved.

**Paper Strengths:**

* The paper demonstrates the use of existing hardware and software techniques for protecting data privacy without compromising features.
* It shows how machine learning at the edge can be practically used to protect data privacy for certain applications.
* The paper clearly elaborates on threat cases, prior art and the necessary features of a trusted execution environment.

**Paper Weaknesses:**

* The pure"wakeword" application allow the reduction of the sensor data to a low data rate bitstream with only non-privacy critical data. Most practical applications however  require at least some pre-processed form of the sensor raw-data which is privacy critical, after the wake event happened (e.g. MFCC data after a wakeword has been spoken). This should be considered. Duplication of sensors as suggested in 3.3 is no real solution for this.



**Poster (If Paper Is Rejected):**

1: Yes, ok for poster sesion to nurture work

**Reviewer Confidence:**

4: The reviewer is confident but not absolutely certain that the evaluation is correct

---

### Official Review · AnonReviewer4 · 2021-01-24

**Overall Merit Score:** 2

**Brief Summary:**

The paper looks into the concerns and potential threat vector of edge devices.  The paper provides threat examples and threat vectors in both the abstract and concrete theoretical examples.

The paper proposes a label (similar to food labels) to explain to users and those installing the device of risks and sensors used.

The paper provides a survey of security and threat mitigations that are available (e.g. Trustzone, TEE, etc.) and introduces some other possible implementations.  The paper also describes some consent models that can be used to allow users to understand the threats that can take place on the edge device.

**Detailed Comments:**

There are many examples in the paper that make assumptions of TinyML devices.  The distinction between TinyML and a broader edge device category is not made.
- In the second column, middle, there is mention of "raw recorded audio, video".  In 2.1, the paper mentions "often store significant amount of data".  This assumptions are typically not true in TinyML devices.  They are more true in typical powered edge devices.  The author does talk about these devices and examples (e.g. smart speakers, video billboards, etc.).

Editorial comment.  In 2.3.1, the author proposes a consent model similar to Prop 65.  Those outside of California may not be familiar with this analogy.  A better term or an explanation may be appropriate for the wider potentially international audience.

The paper does provide many concrete examples of use cases and mitigation.  Those employing edge devices with sufficient memory, power, and communication capabilities may want to refer to these as a good first read for security concerns.

**Paper Strengths:**

The paper provides examples of threat models in both a high level and with concrete examples.  Those who are not familiar with the threat models will be able to quickly understand the threats.  Also, the reader will be able to quickly understand the risk of these threats in a concrete way.

The paper put concrete examples on the amount of memory and some examples of the computation that would be needed with the current available mitigations.

The proposed markings in Figure 1 is something that many devices could benefit from.

**Paper Weaknesses:**

The papers is too high of a level, especially since the paper is not a survey paper.

The papers threat models presented are not necessarily the correct threat models that should be applied for TinyML devices.  For example, most TinyML devices are memory limited and have limited communication capabilities.  These limitation often stem from both a cost consideration and power considerations.  The paper often present threat models that require recording of raw data and the transport of such raw data.  Though these can occur, it would be limited in TinyML use cases because of both memory footprint and power to send these samples out.   For example, the mitigation proposed in Fig 3 is already the model used in most TinyML devices.  This is the model in devices not because of specific security considerations, but because of the power and memory limitations as part of the design.

**Poster (If Paper Is Rejected):**

1: Yes, ok for poster sesion to nurture work

**Reviewer Confidence:**

4: The reviewer is confident but not absolutely certain that the evaluation is correct

---

### Official Review · AnonReviewer2 · 2021-01-28

**Overall Merit Score:** 1

**Brief Summary:**

This paper presents an approach to protect the raw sensor data from unauthorized access by machine learning models, which may lead to serious privacy leakage. A prototype system has been implemented and evaluated.

**Detailed Comments:**

1. Even though a rate-limit policy has been proposed to improve the consent model, it may not be helpful for users to better understand the potential privacy threat.
2. The results of execution time are not well explained. For example, the unit “M” in Table 1 is not clearly defined. In addition, the experimental setup is not clear, e.g., which dataset, whether the results are averaged, etc.
3. In addition to computation efficiency, energy consumption is another important evaluation metric for edge devices. How the proposed design (i.e., computation in TrustZone) will impact the energy consumption is not evaluated in this paper.
4. The writing of this paper can be further improved. For example, ‘always on sensors’->’always-on sensors’, ‘secure world’ and ‘nonsecure world’ are capitalized in some sections but are written in lower case in other sections.

**Paper Strengths:**

1. The privacy issue addressed by this paper is very important, especially for edge devices with ubiquitous on-board sensors. This paper proposes a practical approach to prevent raw sensor data from being maliciously used by machine learning applications.
2. The privacy issues are comprehensively analyzed from different perspectives, including device vulnerabilities, applications, and user consent model. This is very helpful for understanding the potential threats and designing effective defense techniques.
3. A prototype system has been implemented on the real edge device, showing the applicability of the proposed design.


**Paper Weaknesses:**

1. The novelty of this paper limited. The proposed solution is based on existing techniques. It may require some effort for the system implementation, but the research challenges and contributions are not clear.
2. The hardware separation may not be a feasible and practical design, since this requires extra sensors for general purpose. Such requirements are hard to be satisfied in general.


**Poster (If Paper Is Rejected):**

1: Yes, ok for poster sesion to nurture work

**Reviewer Confidence:**

4: The reviewer is confident but not absolutely certain that the evaluation is correct

---

### Official Review · AnonReviewer1 · 2021-01-30

**Overall Merit Score:** 2

**Brief Summary:**

This work presents a practical approach towards hardware security by using a secure enclave to protect sensitive data from attackers. The authors demonstrate that with a simple programming framework, how machine learning application developers can be as productive as usual, while keeping user data private. They also demonstrate our implementation for privacy-preserving machine learning on an embedded system for person-detection.

**Detailed Comments:**

This paper provides a method of privacy-preserving machine learning at the edge. Authors claims that by using a protected enclave, they can shield sensitive sensor data from attackers, and thus only reveal a single bit of output to the rest of the unsecure environment. Authors demonstrated their implementation for the security of privacy on the embedded system by using a reasonably large model for human detection.
The idea is unique where the authors claim that they limit access to the raw sensor data to the applications and only provide limited results from the ML models. To this extent, the use of ARM Trust Zone to provide flexibility to the models is interesting.
However, the work fails to provide sufficient support for a number of its presentations.
•	Even though the authors are targeting implementation on Edge devices, they only provide results for one embedded platform. Along with this, there is no comparison among state-of-the-art security protocols of similar kinds. This makes it difficult to create a complete picture with regards to the efficacy of the work.
•	The authors use Coral Camera and Google Coral board to process the initial signals but do not provide detail descriptions of these processes anywhere. Also, section 5 fails to provide convincing write up explaining the flow of the secure enclave. This needs to be clearer with detailed block diagrams.
•	The overall frame of the paper is inconsistent, making it very difficult to understand what the actual objectives and contributions are.


**Paper Strengths:**

1.	Unique approach towards providing hardware security.
2.	Ensuring Flexibility to deal with large ML models.


**Paper Weaknesses:**

1.	The paper seems to be in its early stages of progress. There is no evaluation and performance analysis, not implementation. They talk about different scenarios but there is no presentation for that. contribution.
2.	It is difficult for the reviewer to understand the contribution in TinyML domain.
3.	The paper does not present any details for their hardware architectures.
4.	The paper is weak documented. Figures are of low qualities. Organization of the paper should be changed as well. The flow of the paper is very inconsistent.


**Poster (If Paper Is Rejected):**

1: No, paper is below bar for poster as well

**Reviewer Confidence:**

5: The reviewer is absolutely certain that the evaluation is correct and very familiar with the relevant literature

---

### Decision · Program_Chairs · 2021-02-05

**Decision:**

Accept (Poster)

**Comment:**

Based on the reviewer feedback, your paper has been accepted as a poster.

Please read the reviews carefully and make sure the concerns are addressed in your poster submission.

Accepted posters are given a 5-minute slot for an oral presentation on Friday, March 26, 2021, to pitch the main ideas of your work and to stimulate discussions. Detailed instructions will follow soon. All final posters will earn a stamp of acceptance as such: “Published as a poster at TinyML Research Symposium 2021.”